# Adjusting the Connection Length of Additively Manufactured Electrodes Changes the Electrochemical and Electroanalytical Performance

**DOI:** 10.3390/s22239521

**Published:** 2022-12-06

**Authors:** Robert D. Crapnell, Alejandro Garcia-Miranda Ferrari, Matthew J. Whittingham, Evelyn Sigley, Nicholas J. Hurst, Edmund M. Keefe, Craig E. Banks

**Affiliations:** Faculty of Science and Engineering, Manchester Metropolitan University, Chester Street, Manchester M1 5GD, UK

**Keywords:** additive manufacturing, electrochemistry, electrodes, 3D-printing, electroanalysis

## Abstract

Changing the connection length of an additively manufactured electrode (AME) has a significant impact on the electrochemical and electroanalytical response of the system. In the literature, many electrochemical platforms have been produced using additive manufacturing with great variations in how the AME itself is described. It is seen that when measuring the near-ideal outer-sphere redox probe hexaamineruthenium (III) chloride (RuHex), decreasing the AME connection length enhances the heterogeneous electrochemical transfer (HET) rate constant ($k^{0}$) for the system. At slow scan rates, there is a clear change in the peak-to-peak separation (Δ*Ep*) observed in the RuHex voltammograms, with the Δ*Ep* shifting from 118 ± 5 mV to 291 ± 27 mV for the 10 and 100 mm electrodes, respectively. For the electroanalytical determination of dopamine, no significant difference is noticed at low concentrations between 10- and 100-mm connection length AMEs. However, at concentrations of 1 mM dopamine, the peak oxidation is shifted to significantly higher potentials as the AME connection length is increased, with a shift of 150 mV measured. It is recommended that in future work, all AME dimensions, not just the working electrode head size, is reported along with the resistance measured through electrochemical impedance spectroscopy to allow for appropriate comparisons with other reports in the literature. To produce the best additively manufactured electrochemical systems in the future, researchers should endeavor to use the shortest AME connection lengths that are viable for their designs.

## 1. Introduction

Additive manufacturing (AM, 3D-printing) is a technology utilized to build 3-dimensional objects in a layer-by-layer fashion from computer-generated design files. It has transcended many industrial categories, with various research groups across the globe pushing the boundaries of 3D and 4D printed items [1]. The field of AM has rapidly expanded from the invention of stereolithography (SLA) in 1983 to incorporate many different printing methodologies, such as Direct Metal Laser Sintering (DMLS) and Fused Filament Fabrication (FFF, commonly referred to a Fused Deposition Modeling or FDM) [2]. Research into FFF has become increasingly popular due to its rapid prototyping capabilities, low wastage, and the continuing reduction in cost of entry [3]. The process works through the initial creation of a CAD (computer-aided design) model of the desired item, which can be converted into a “.GCODE” file for the printer. The printer can then follow this code, extruding the loaded thermoplastic filament into the shape of the design in a layer-by-layer fashion.

It is the progression of FFF, including the development of commercial conductive filaments, that has helped to cement the use of AM within the field of electrochemistry and energy storage [4,5,6,7]. The vast majority of research in this field has focused on using two commercial conductive filaments (BlackMagic and Protopasta), both using poly(lactic acid) (PLA) as the base polymer, but with either graphene or carbon black incorporated as active material additives to provide the conductivity. These were traditionally printed into discs or a “lollipop” shape, mimicking the 2D image used for most screen-printed electrodes [8], with similar dimensions and used for electrochemical applications. The initial efforts of researchers were put into the “activation” of printed parts to improve their electrochemical performance [9]. There is a vast array of published methods to achieve this activation, such as simple mechanical polishing [10], electrochemical scanning [11,12,13], submersion/sonication in solvents [14] or reducing agents [15], carbonization [16], thermal annealing [17], or laser-scribing [18]. These methods all typically revolve around the removal of parts of the PLA matrix, revealing increased amounts of conductive material and producing the desired enhanced electrochemical performance of the additively manufactured electrode (AME). Each of these methods is more suited for certain applications and set-ups, as discussed by Rocha and co-workers [9].

Recently, electrochemical platforms have been designed that allow for physically defined electrode areas to be produced in a single print [19]. This important step allows for increased reproducibility in the production of AMEs and exhibits how full sensing platforms can be produced in a single print using independent dual-extruder printers [20]. There are many other AM electrochemical platforms reported in the literature utilizing multiple prints or post-print assembly [11,21,22,23,24,25,26], with little explanation given to the cell design or the AMEs themselves. Only recently have the printing or physical parameters of the AME begun to be explored, with work showing that thinner layer thickness [27], vertical printing orientation [27,28], increased printing temperature [29], and most recently, the age of the filament/print [30] can produce increased conductivity in printed parts. These works show enhancement in the performance of parts by only altering one or two parameters to do with the manufacture of the part.

In this manuscript, we vary the physical connection length of AMEs, shown recently for screen-printed electrodes [31], and explore how this can affect electrochemical performance. We highlight how this key parameter affects various electrochemical applications of AMEs. This is of significant importance to the field of AM in electrochemistry and highlights how the best possible electrochemical platforms with be achieved in the future.

## 2. Experimental Section

### 2.1. Materials

All chemicals used were of analytical grade and were used as received without any further purification. All solutions were prepared with deionized water of resistivity not less than 18.2 MΩ cm. Hexaamineruthenium (III) chloride (RuHex), sodium hydroxide, dopamine hydrochloride, and phosphate-buffered saline (PBS) tablets were purchased from Merck (Gillingham, UK). Potassium chloride was purchased from Fisher Scientific (Loughborough, UK). The conductive PLA/carbon black (PLA/CB) filament (1.75 mm, Protopasta) and non-conductive PLA filament (Red 1.75 mm, Raise3D, Irvine, CA, USA) were purchased from Farnell (Leeds, UK).

### 2.2. Additive Manufacturing

The defined AMEs were produced using FFF, Figure 1A, on a Raise3D E2 Independent dual-extruder (IDEX) 3D-printer (Raise3D, USA) using conductive PLA/carbon black filament (1.75 mm) on the right extruder and non-conductive PLA (Raise3D, USA) on the left. All designs and “.3MF” files were produced using Autodesk^®^ Fusion 360^®^, then sliced and converted to “.GCODE” files using the printer-specific software IdeaMaker. The printing parameters used in this work were identical for each AME, these were a nozzle temperature of 210 °C, bed temperature of 60 °C, layer height of 0.2 mm, layer width of 0.4 mm, and 100% infill on the conductive PLA/CB filament.

### 2.3. Electrochemical Experiments

An Autolab PGSTAT128N potentiostat (Utrecht, The Netherlands) was used in conjunction with NOVA 2.1.5 (Utrecht, The Netherlands) to carry out electrochemical measurements using a three-electrode configuration. The AMEs were used as the working electrodes and connected to the potentiostat via an edge connector (RS Components, Corby, UK), a nichrome wire coil was used as the counter electrode, and an Ag|AgCl electrode was used as the reference in all cases. All solutions were prepared using deionized water of resistivity not less than 18.2 MΩ cm from a Milli-Q system (Merck, Gillingham, UK). Solutions of RuHex were degassed thoroughly for at least 15 min with nitrogen prior to any electrochemical measurement.

Activation of the AMEs was performed before all electroanalytical experiments. This was achieved electrochemically in NaOH, as described in the literature [12]. Briefly, the AMEs were connected as the working electrode in conjunction with a nichrome wire coil counter and Ag|AgCl reference electrode and placed in a solution of NaOH (0.5 M). Chronoamperometry was used to activate the AME by applying a set voltage of +1.4 V for 200 s, followed by applying—1.0 V for 200 s. The AMEs were then thoroughly rinsed with deionized water and dried under compressed air before further use.

Scan rate studies (5–1000 mV/s) were performed using cyclic voltammetry from +0.3 V to −0.7 V against an Ag|AgCl reference electrode with a step potential of 0.00244 V. Electrochemical impedance spectroscopy was performed at −0.14 V for experiments using RuHex from 100,000 to 0.1 Hz at 10 frequencies per decade with an amplitude of 0.01 V.

Calculations of the Heterogeneous Electron Transfer (HET) rate constant were performed using the peak-to-peak separation (Δ*E_p_*) to deduce the kinetic parameter (*ψ*)*,* where Δ*E_p_* is obtained at various voltammetric scan rates. The standard heterogeneous constant ($k^{0}$) can be calculated via the gradient when plotting *ψ* against [*πDn*ν*F*/*RT*]^−1/2^. In cases where ΔE_p_ is bigger than 212 mV, the following equation should be implemented: $k_{obs}^{0}=[\;2.18{\left(\frac{\alpha Dn\nu F}{RT}\right)}^{-\frac{1}{2}}\exp [-\left(\frac{\alpha nF}{RT}\right)\Delta Ep]\;$ where α is assumed to be 0.5. We use this approach as this is the most commonly adopted approach in the academic literature to determine the HET [32].

## 3. Results and Discussion

The development of AMEs throughout the literature is dominated by the use of commercially available conductive filaments, such as the PLA/carbon black (PLA/CB) based filament Protopasta (PP). One of the advantages of AM is the ability to design and manipulate a vast array of shapes and sizes simply using CAD, tailoring every parameter to the desired application. Even so, the vast majority of publications using AM for electrochemical applications use the same “lollipop” or disc electrode shape. A summary of these can be found in Table 1, highlighting the similarities in the works, where many researchers use identical electrodes and others change only the physical dimensions of the disc. This is often done with very little explanation, in some cases, the dimensions are not even reported, even though these alterations could have a dramatic effect on the electrochemical performance. This means direct comparison between reported electrochemical platforms is impossible. Herein, we look to show how simply changing the connection length of an AME affects its performance in various electrochemical applications.

### 3.1. Effect on the Electrochemical Performance

The AMEs used in this work were produced using FFF printing, Figure 1A, on an IDEX 3D-printer. This allowed for the design and production of physically defined 3.1 mm disc AMEs (mimicking a screen-printed electrode) [31] of differing connection lengths (10, 25, 50, 75, and 100 mm), Figure 1B, where the right extruder deposited the conductive PLA/CB filament, and the left extruder deposited a non-conductive PLA, which acted as a casing. This casing allowed for the production of AMEs with defined working electrode areas, independent of researcher placing [19,20]. This is a significant improvement over the use of lollipop electrodes as it can act as a disc electrode rather than a cylinder [19]. With lollipop electrodes, the two discs can produce significantly different electrochemical performances whether they are printed directly onto the print bed or the top layer of the print [33]. Additionally, the electrochemically active area of the electrode is entirely dependent on how far the researcher places the lollipop into the solution, which will almost certainly change when changing electrodes, and even when placing the same electrode into different solutions. Utilizing the IDEX printing, the electrochemical set-up is significantly more reproducible, Figure 1C, where there are external layers of non-conductive material between the working electrode and its connection point to the edge connector. This ensures the same electrochemically active area is in contact with the solution (whatever that may be) in all cases. We note that all AMEs were activated using identical conditions prior to any experiments [12].

**Table 1 sensors-22-09521-t001:** A table comparing different FFF AMEs found in the literature. Comparing the shape, size, thickness, connection length, and the application of the AME.

Conductive Filament	Shape	Size	Thickness	Connection Length	Application	Reference
PLA/CB (PP)	Disc	D: 5 mm	1 mm	34 mm	Electroanalytical detection of NBOMes	[11]
PLA/CB (PP)	Disc	D: 54 mm	N/A	N/A	Electroanalytical detection of dopamine	[12]
PLA/NG	Disc	D: 3 mm	1 mm	34 mm	Electroanalytical detection of Mn	[13]
PLA/nanocarbon	Disc	D: 11 mm	0.3 mm	N/A	Electrochemical capacitors	[16]
PLA/CB (PP)	Disc	D: 5.4 mm	N/A	N/A	Electroanalytical detection of Cd, Pb and Cu	[18]
PLA/CB (PP)	Disc	D: 3 mm	0 mm (Defined)	18 mm	Electroanalytical detection of ascorbic acid and acetaminophen	[20]
PLA/G (BM)	Lollipop	D: 5 mm	1 mm	20 mm	Electroanalytical detection of L-methionine	[22]
PLA/CB (PP)	Lollipop	D: ~5 mm	N/A	~20 mm	Electroanalytical detection of Hg, glucose, caffeine	[23]
PLA/G (BM)	Disc	D: 15 mm	2 mm	N/A	Electroanalytical detection of picric acid	[34]
PLA/G (BM)	Lollipop	D: 5 mm	2 mm	20 mm	Oxygen evolution reaction	[35]
PLA/G (BM)	Disc	D: 15 mm	2 mm	N/A	Electroanalytical detection of 1-naphthol	[36]
PLA/G (BM)	Disc	D: ~5 mm	N/A	~40 mm	*Pseudo*-reference electrode	[37]
PLA/G (BM)	Disc	D: 3 mm	1 mm	N/A	Electrochemical energy storage	[38]
PLA/NG	Lollipop	D: 3 mm	1 mm	N/A	Electroanalytical detection of Pb and Cd	[39]
PLA/CB/2D-MoSe_2_	Lollipop	D: 3.5 mm	1.5 mm	10 mm	Electrochemical water splitting	[40]
PLA/G (BM)	Lollipop	D: 8 mm	1.6 mm	45 mm	Electroanalytical detection of mycotoxin zearalenone	[41]
PLA/G (BM)	Lollipop	D: 8 mm	2 mm	37 mm	Electroanalytical detection of hydrogen peroxide	[42]
PLA/G (BM)	Lollipop	D: 6 mm	1 mm	N/A	Electrode benchmarking	[43]
PLA/G (BM)	Lollipop	D: 5 mm	1 mm	“few cm”	Hydrogen evolution reaction	[44]
PLA/CB (PP)	Lollipop	D: 5 mm	1 mm	50 mm	Hydrogen evolution reaction	[45]
PLA/G (BM)	Disc	D: 4.8 mm	N/A	N/A	Electroanalytical detection of cocaine	[46]
PLA/CB (PP)	Disc	D: 3.8 mm	N/A	N/A	Electroanalytical detection of 2,4,6-trinitrotoluene	[47]
PLA/CB (PP)	Lollipop	D: ~4 mm	N/A	~11 mm	Electroanalytical detection of glucose	[48]
PLA/CB (PP)	Disc	D: 5 mm	N/A	N/A	Electroanalytical detection of forensic drugs	[49]
PLA/G (BM)	Lollipop	D: 6 mm	N/A	N/A	Electroanalytical detection of COVID-19	[50]
PLA/CB (PP)	Disc	D: 2.85 mm	10 mm	N/A	Electroanalytical detection of Hantavirus Araucaria nucleoprotein	[51]
PLA/CB (PP)	Lollipop	D: 5 mm	2 mm	25 mm	Effect of water ingress on AMEs	[52]
PLA/CB (PP)	Disc	D: 4 mm	N/A	~20 mm	Electroanalytical Sensors	[53]
PLA/CB (PP)	Disc	D: 3.55 mm	N/A	N/A	Electroanalytical detection of adrenaline	[54]
PLA/CB (PP)	Disc	D: 5.4 mm	1.8 mm	40 mm	Electroanalytical detection of naproxen	[55]
PLA/G (BM)	Lollipop	D: 6 mm	N/A	20 mm	Electroanalytical detection of L-cysteine	[56]

Key: PLA/G: Polylactic acid/graphene; BM: Black Magic 3D commercial filament; PLA/CB: Polylactic acid/carbon black; PP: Protopasta; PLA/NG: Polylactic acid/nanographite D: Diameter; NBOMes: *N*-benzylmethoxy-derivatives.

Firstly, it is useful to know how much resistance is introduced into the electrochemical set-up through the working electrode itself [31]. This can be achieved primarily through two methods: (1) Through simple multimeter measurements between the working electrode and the connection point or, (2) through electrochemical impedance spectroscopy (EIS), where the value at the beginning of the semi-circular Nyquist plot is an indicator of the resistance in the experimental set-up. The resistance values measured utilizing both these methodologies are presented in Table 2, where both values show good agreement in trend, although slight deviations in absolute value. Figure 2A presents Nyquist plots for the five studied lengths of AME against hexaamineruthenium chloride (RuHex, 1 mM, −0.14 V), which clearly shows the characteristic semi-circular plot. The starting point for these plots shifts to larger real impedance (Z’) values with increasing connection lengths, indicating an increase in the resistance. It can be seen that the charge transfer resistance (R_CT_) for each system remains similar in magnitude, which is expected for measurements using the same working electrode shapes in identical solutions. This indicates that the variations are solely caused by changes in the connection length of the AME. Additionally, it can be seen that the error obtained through using EIS is significantly lower, and we, therefore, recommend this technique for the measurement of AME resistance where possible. Referring back to Table 2, it can be seen that the resistance measurement across the length of the AMEs increased significantly from 1.24 ± 0.03 kΩ to 9.00 ± 0.18 kΩ for the 10- and 100-mm connection lengths, respectively. This highlights the wide variation in AME resistances that can be produced through different designs, and this parameter should, therefore, always be reported. The changes in resistance values within the working electrode’s circuit resistance (between the working electrode and its connections) observed herein are likely due to the use of graphitic active materials within inter binders, similar to those observed within graphitic inks [31]. This, however, should not happen when metallic electrode connections are used.

To see how this change in the overall electrode resistance affects the electrochemical performance of the AMEs, scan rate studies (5–1000 mV s^−1^) were performed for five electrodes of each length against the near-ideal outer-sphere redox probe RuHex (1 mM in 0.1 M KCl). An example of the response obtained is presented in Figure 2B for an AME with 10 mm connection length. Figure 2C shows a representation of the cyclic voltammetric response for an AME of each length at a set scan rate of 50 mV s^−1^, and the corresponding heterogeneous electrochemical transfer (HET) rate constant (*k*^0^) for the system is also presented as a function of the AME length (Figure 2C). From Figure 2C, it can be seen that even at slow scan rates such as this, there is a clear change in the peak-to-peak separation (Δ*Ep*) observed, with the Δ*Ep* shifting from 118 ± 5 mV to 291 ± 27 mV for the 10 mm and 100 mm electrode respectively. This is also echoed by the observation of the HET, which is presented as the length of the AME, which appears to be linear. Thus, this corresponds to a massive shift in the electrochemical performance of the AME, in addition to a large drop in reproducibility. It can be seen on this plot that there is some variation in the observed peak currents, which is expected for AMEs due to variations in the printing between electrodes, but this is not a consistent trend with varying connection lengths. This can have an effect on both the peak current, Figure 3A, and calculated electrochemical active area. Figure 3B shows where some variation can be seen across the different connection lengths.

This section has shown the dramatic effect that changing the physical parameters of AMEs can have on the electrochemical performance of the system. In summary, it is shown herein that the use of shorter graphitic connections for AMEs results in a more accurate determination of the HET (or how longer connections can have an effect on this). We next turn our attention to looking at the effect changing the connection length has on the electroanalytical performance of AMEs.

### 3.2. Effect on the Electroanalytical Performance

From inspection of Table 1, it can be seen that a plethora of publications have reported the production of electroanalytical sensing platforms utilizing AM. We have seen from the previous section that changing the connection length of AMEs can have a significant effect on the measured electrochemical response of the system. Following this, we look toward the electroanalytical response of AMEs. Figure 4A presents cyclic voltammograms for the detection of dopamine (5–200 µM) using a 50 mm connection length AME. This shows the characteristic oxidation peak for the conversion of dopamine-to-dopamine quinone. Figure 4B shows the calibration curves obtained for the detection of dopamine (5–200 µM) using 10-, 50-, and 100-mm connection length AMEs. It can be seen that at these concentrations, there was no significant difference obtained between the three lengths studied, Table 3. The 10-, 50-, and 100-mm systems obtained Limits Of Detection (LOD) of 1.19, 1.73, and 1.55 µM and sensitivities of 18.8, 18.2, and 17.5 nA µM^−1^, respectively. Figure 4C presents voltammograms for the detection of 50 µM dopamine, highlighting the similarities in the obtained cyclic voltammetric response. This shows good agreement with work shown previously for screen-printed electrodes, where at low concentrations, the obtained voltammograms showed good agreement with different connection lengths, with deviations happening at higher concentrations of the analyte [31]. It should be noted that the differences in the peak currents obtained in Figure 4C are due to natural variations in the electrodes and, as seen above in Figure 3, the connection length does not have a significant effect on the peak current for these concentrations. To explore this, higher concentrations of dopamine were tested with the detection of 1 mM dopamine, presented in Figure 4D. It can be seen that at this increased concentration, the peak oxidation potential for dopamine begins to shift to higher potentials as the connection length of the AME is increased. For a 10 mm connection length, the oxidation potential was found to be at +0.35 V compared to +0.42 V for a 50 mm connection length and +0.50 V for a 100 mm connection length versus an Ag|AgCl reference, showing a 150-mV shift in peak oxidation potential solely due to increasing the connection length of an AME by 10 times. This is because as the concentration of the analyte is increased, their associated current response increases so the IR drops. However, when lower concentrations (and currents) are used, with the lesser IR drop, the voltammograms are less distorted and behave similarly to a zero resistance case [31]. This highlights the importance of reporting AME lengths and resistances in future publications.

This section has provided evidence to show that at low concentrations, the connection length does not appear to significantly influence the sensing capabilities of AMEs. However, at higher concentrations, the peak potentials can shift by huge amounts due to changes in the AME connection length. Therefore, the most reliable AM electroanalytical sensing platforms over a wide concentration range will be produced using the shortest electrode connection length possible. As such, this should be incorporated into the design of future sensing platforms and reported as such.

## 4. Conclusions

In this work, the vital physical parameter of connection length of AMEs is explored toward electrochemical applications. By decreasing the connection length, the resistance measured through the AME is reduced significantly. The resistance of an AME can be measured through multimeter measurements or EIS, with the latter providing more reproducible results. It is seen that increasing the connection length of AMEs significantly reduces the electrochemical rate constant for the system, whilst smaller variations in the electrochemical surface areas and peak currents are a result of the printing process. At slow scan rates, there is a clear change in the peak-to-peak separation (Δ*E_p_*) observed in the RuHex voltammograms, with the Δ*E_p_* shifting from 118 ± 5 to 291 ± 27 mV for the 10 and 100 mm electrode, respectively. When used for the electroanalytical determination of dopamine, at low concentrations, there was little variation obtained between the AMEs of different connection lengths. However, at high concentrations (1 mM), the peak oxidation potential was shifted significantly, with a shift of 150 mV between the 10 and 100 mm connection lengths. This work highlights how the reporting of AME connection lengths and AME resistance values are of vital importance for the true comparison between systems in the literature. Additionally, to produce the best electroanalytical sensing platform with the widest linear range, an AME with the shortest viable connection length should be used.

## Figures and Tables

**Figure 1 sensors-22-09521-f001:**
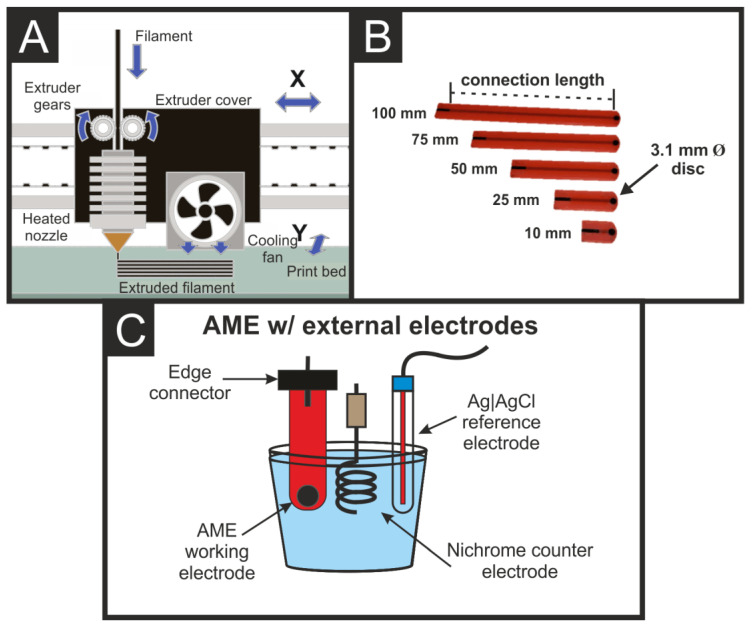
(**A**) Schematic representation of the printer head for an FFF printer, highlighting the different parts and layer-on-layer print of the extruded filament. (**B**) Photograph of the printed 3.1 mm diameter AMEs with different connection lengths (100, 75, 50, 25, and 10 mm). (**C**) Schematic of the experimental set-up used throughout this work, utilizing an additively manufactured working electrode connected through an edge connector, Ni coil counter electrode, and Ag|AgCl reference electrode.

**Figure 2 sensors-22-09521-f002:**
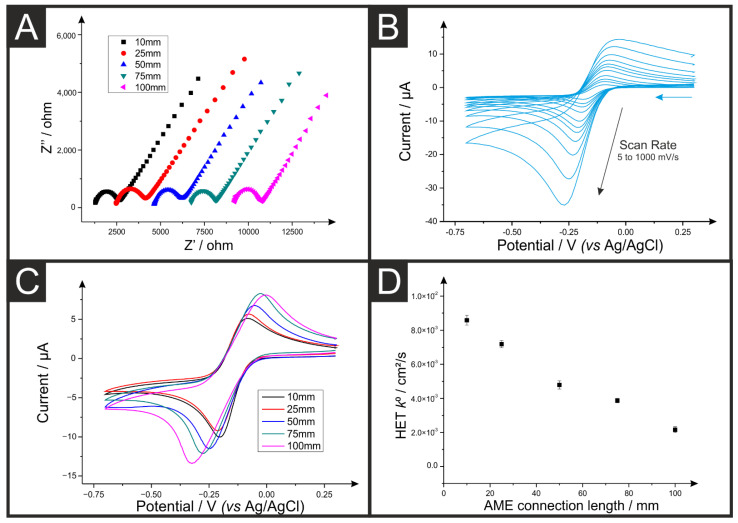
(**A**) Nyquist plot of hexaamineruthenium (III) chloride (1 mM, 0.1 M KCl), performed using an AME (10, 25, 50, 75, and 100 mm connection lengths) produced from PLA/CB, nichrome coil counter electrode and Ag|AgCl reference electrode. (**B**) Cyclic voltammograms (5, 10, 15, 25, 50, 75, 100, 200, 300, 500, and 1000 mV s^−1^) of hexaamineruthenium (III) chloride (1 mM, 0.1 M KCl), performed using an AME (10 mm connection length) produced from PLA/CB, nichrome coil counter electrode and Ag|AgCl reference electrode. (**C**) Cyclic voltammograms (50 mV s^−1^) of hexaamineruthenium (III) chloride (1 mM, 0.1 M KCl), performed using an AME (10, 25, 50, 75, and 100 mm connection lengths). (**D**) Plot of the calculated electrochemical rate constants (*k*^0^) for AMEs of different connection lengths.

**Figure 3 sensors-22-09521-f003:**
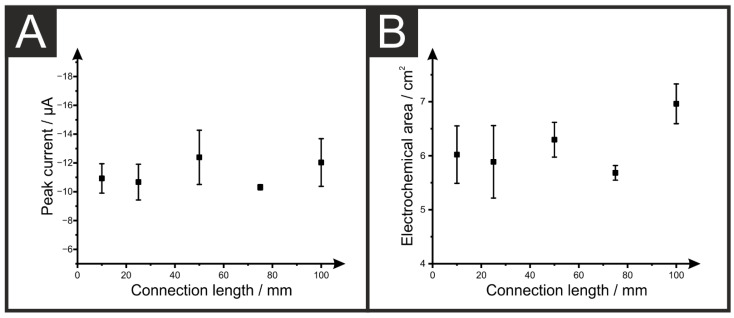
(**A**) Plot of the average peak current (*n* = 5) obtained for the reduction of RuHex (1 mM in 0.1 M KCl) at 50 mV s^−1^ for AMEs of different connection lengths. (**B**) Plot of the average calculated electrochemical area of the different connection length AMEs.

**Figure 4 sensors-22-09521-f004:**
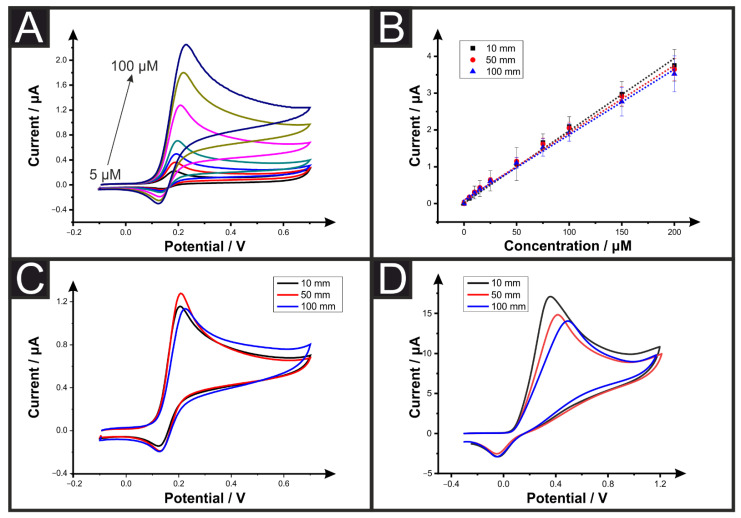
(**A**) Electroanalytical detection of dopamine (5, 10, 15, 25, 50, 75, 100 µM) using cyclic voltammetry (scan rate = 0.05 V s^−1^) using a 50 mm connection length AME with nichrome wire counter electrode and Ag|AgCl reference electrode. (**B**) Calibration plots for the detection of dopamine (5–200 µM) using cyclic voltammetry (scan rate = 0.05 V s^−1^) using 10-, 50-, and 100-mm connection length AMEs. (**C**) Comparison of the cyclic voltammograms (scan rate = 0.05 V s^−1^) for the detection of dopamine (50 µM) using 10-, 50-, and 100-mm connection length AMEs. (**D**) Comparison of the cyclic voltammograms (scan rate = 0.05 V s^−1^) for the detection of dopamine (1 mM) using 10-, 50-, and 100,mm connection length AMEs.

**Table 2 sensors-22-09521-t002:** Effect of changing the connection length on the resistance measured through a multimeter and electrochemical impedance spectroscopy.

Electrode Connection Length (mm)	Multimeter Resistance (*n* = 12, kΩ)	EIS Resistance (*n* = 5, kΩ)
10	1.32 ± 0.20	1.24 ± 0.03
25	2.65 ± 0.11	2.55 ± 0.06
50	4.85 ± 0.12	4.67 ± 0.08
75	7.18 ± 0.33	6.81 ± 0.07
100	9.07 ± 0.13	9.00 ± 0.18

**Table 3 sensors-22-09521-t003:** A table comparing electroanalytical response of AMEs with different connection lengths to the detection of dopamine (5–200 µM) using cyclic voltammetry (50 mV s^−1^).

AME Connection Length (mm)	R^2^	Sensitivity (µA µM^−1^)	LOD (µM)
10	0.9957	0.0188	1.19
50	0.9945	0.0182	1.73
100	0.9959	0.0175	1.55

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
