# Peer review of "Adjusting the Connection Length of Additively Manufactured Electrodes Changes the Electrochemical and Electroanalytical Performance"

_sensors, 2022, doi:10.3390/s22239521_

Round 1
Reviewer 1 Report
In this article, the authors addressed one of the critical points for the additively manufactured electrode (AME) that plays important role in the electrochemical performance and sensing while used as current collector. It is true that in recent times several studies had been published using AME as current collector or electrode overlooking the connector length. The article can be published after corrections addressing following comments.
1. It is clear from the EIS study that with increasing the connecting length inherent resistance of the electrode increases. The author could address in the discussion section, the reason why such inherent resistance increased with connection length.
2. For Figure 2B, please mention individual scan rates.
3. The authors should address why there is different in behavior between low concentration and high concentration of dopamine with the connection length. As the increase of connection length increase the inherent resistance of the electrode, how the results are affected by the high concentration of electrolyte, need to be explained.
4. Figure 4C shows that 50 mm of connecting length has higher current response (peak value) which is not following the trend of connecting length to current response. However, in case of high concentration Figure 4D, although there is peak shift but it follows the trend. Authors should clearly explain both cases.
5. The authors have cited several articles those using the AME as current collector. The authors also could add these articles in the Introduction section.
a) Nanoscale 13 (11), 5744-5756
b) Chemistry–A European Journal 26 (67), 15746-15753
6. In Figure 3A, please correct the typo “KCL”.
Reviewer 2 Report
My general evaluation for the article titled “Adjusting the connection length of additively manufactured electrodes changes the electrochemical and electroanalytical performance” is as follows. It is a good study done in the field of additive manufacturing. It seems to have been edited and written in accordance with the purpose of the work performed. It is believed that the following corrections will be beneficial for the strengthening of the article. My assessment for the article is as follows:

Reviewer 3 Report
The paper 'Adjusting the connection length of additively manufactured electrodes changes the electrochemical and electroanalytical performance' by Crapnell R. is reporting the electrochemical and electroanlaytical performance of additive manufactured electrode. The resistance of the connection lengths is an important parameter and it has been assessed from an electrochemical approach (with RuHex) and electroanalytical approach (dopamine).
I would like you to adress the following points
-Figure 3A what about the peak to peak separation? Is there something you can conclude from it? The reported difference in peak current seems to be neglegible(within the experimental error)
-Figure 3B. I do not get the influence of the connection length with the electrochemical area. The response is quite flat and I am struggling to see a relationship between those two parameters
-The relationship between the con-ection lengths at the resistance almost scales of a factor of 10 but what is the relationship between the limit of detection (dopamine) and the connection lengths?
-Depending on the pH dopamine could be more prone to polymerization, this could explain the shift of the peak of Figure 4D. Could you please elaborate?
Best Regards,
Round 2
Reviewer 3 Report
Thanks for addressing my comments. Best Regards